# Peptidyl Arginine Deiminase 2 (PADI2)-Mediated Arginine Citrullination Modulates Transcription in Cancer

**DOI:** 10.3390/ijms21041351

**Published:** 2020-02-17

**Authors:** Miguel Beato, Priyanka Sharma

**Affiliations:** 1Gene Regulation, Stem Cells and Cancer Program, Centre for Genomic Regulation (CRG), The Barcelona Institute of Science and Technology (BIST), Dr. Aiguader 88, 08003 Barcelona, Spain; 2Universitat Pompeu Fabra (UPF), 08003 Barcelona, Spain

**Keywords:** arginine citrullination, RNA polymerase II, transcription, pause release, cell proliferation, P-TEFb complex, histone H3, chromatin, phase separation, cancer

## Abstract

Protein arginine deimination leading to the non-coded amino acid citrulline remains a key question in the field of post-translational modifications ever since its discovery by Rogers and Simmonds in 1958. Citrullination is catalyzed by a family of enzymes called peptidyl arginine deiminases (PADIs). Initially, increased citrullination was associated with autoimmune diseases, including rheumatoid arthritis and multiple sclerosis, as well as other neurological disorders and multiple types of cancer. During the last decade, research efforts have focused on how citrullination contributes to disease pathogenesis by modulating epigenetic events, pluripotency, immunity and transcriptional regulation. However, our knowledge regarding the functional implications of citrullination remains quite limited, so we still do not completely understand its role in physiological and pathological conditions. Here, we review the recently discovered functions of *PADI2*-mediated citrullination of the C-terminal domain of RNA polymerase II in transcriptional regulation in breast cancer cells and the proposed mechanisms to reshape the transcription regulatory network that promotes cancer progression.

## 1. Introduction

Deimination or citrullination as first described in 1958 by Rogers and Simmonds is a post-translational conversion of peptidyl-arginine to non-coded peptidyl-citrulline catalyzed by Ca^2+^- dependent peptidyl arginine deiminase (PADI) enzymes [1,2,3,4]. Citrullination produces a loss of a positive charge, which increases the mass by 0.984 Da [5], and the acidity of the amino acid side chain, as the iso-electric point (pI) changes from 11.41 for arginine to 5.91 for citrulline [6]. Studies in human embryonic kidney (HEK-293) cells found hundreds of proteins that can undergo citrullination, in which 25% of total citrullinated sites contained arginine-glycine-glycine (RGG)/arginine-glycine (RG) binding motifs [7]. These results highlighted that RNA-binding proteins can undergo citrullination to modulate the RNA processing. Citrullination also increases the hydrophobicity and conformation of proteins, which can potentially affect hydrogen bond formation, protein structure, protein-protein interactions and protein-nucleic acid interactions [7,8,9]. Given the significance of these interactions for essential cellular functions, citrullination may uniquely modulate cellular processes (Figure 1). The transformation in protein complexes can change the half-life, stability, and also expose proteins to degradation. For example, citrullination of filament aggregating protein that binds to keratin fibers in epithelial cells (filaggrin) leads to the partial unfolding of this protein, decreases their affinity for keratins, and, thereby, makes it more prone to degradation by several proteases including caspase 14 [10,11]. Additional evidence that citrullination affects the functional protein-protein interactions derives from observation in macrophage differentiation, where citrullination of plasminogen activator inhibitor-2 inhibits its binding ability to proteasome subunit beta type-1 to then modulate the inflammatory response [12].

Changes in citrullination occur in several pathological conditions such as autoimmune disorders and chronic inflammation-related diseases, including periodontitis, rheumatoid arthritis, atherosclerosis, and diabetes [13,14,15] (see Alghamdi, M. et al. [13] for a detailed review on the intrinsic role of citrullination in autoimmune disorders and Olsen, I. et al., [14] for a detailed review on citrullination as a plausible link to periodontitis, rheumatoid arthritis, atherosclerosis, and Alzheimer’s disease). Altered levels of citrullination have also been shown in neurological disorders and prion diseases, [16,17] and respiratory disorders [18,19]. Citrullination can generate new epitopes, which can produce new autoantigens linked to altered immune responses (see [20] for a review on citrullination and autoimmunity). The citrullination of histones in neutrophils facilitates neutrophil extracellular trap (NET) formation or NETosis, which is implicated in innate immune responses ([21]; see [22] for a detailed review on citrullination and NET). Remarkably, citrullination-directed chromatin decondensation leads to transcriptional firing in neutrophils that facilitates NET formation [23]. Moreover, recent work demonstrated that citrullination in macrophages induced inflammatory processes or the pyroptotic form of cell death, which is required for inflammasome assembly and proinflammatory interleukin-1β (IL-1β) release [24]. Moreover, citrullination contributes to cancer progression. Recent research seeks to elucidate how citrullination of several essential proteins drives multiple disease conditions. In this review, we focus on the role of PADI2-mediated arginine citrullination and their functional impact on transcriptional regulation and cancer progression.

## 2. Functional Role of PADI Family

Rogers and Taylor [25] discovered a previously unidentified enzymatic activity when generating peptidyl-citrulline in hair follicle extracts in 1977. The responsible enzymes were later identified as peptidyl arginine deiminases (*PADI*). The *PADI* family is a group of five calcium-dependent enzymes, *PADI1*, *PADI2*, *PADI3*, *PADI4*, and *PADI6*, encoded by the *PADI* loci in human [26,27] (Figure 2A). Among family members, *PADI4* contains a canonical nuclear localization signal (NLS) and hence was long believed to be only *PADI* that can be localized to the nucleus [3,28]. Recently, work by the Thompson group had shown that *PAD2* translocates into the nucleus in response to calcium signaling. They found that binding of calcium to *PADI2* switches its binding from annexin 5 to the RanGTPase in the cytoplasm that promotes translocation of *PAD2* into the nucleus [29]. Using a calcium-dependent process, a series of conformational changes eventually produce the correct movement of key active residues, including a free cysteine 645 into positions that are competent for catalysis [30]. *PADI4* binds five calcium ions per monomer, and calcium binding upregulates enzymatic activity at least by 10, 000-fold without directly participating in catalysis [31]. In *PADI2*, calcium binding occurs in an ordered fashion at six calcium binding sites and induces a calcium switch (made of calcium binding sites 3, 4, and 5) that controls the overall calcium dependence of the enzyme [32]. Another important component for efficient *PADI2* activity is the presence of a reducing environment that maintains the active site cysteine 645 free for catalysis of citrullination [26,27,33].

Human *PADI* genes are localized together as a cluster in a single topological associated domain (TAD) on chromosome 1 (Figure 2A), but most *PADI* gene members exhibit tissue-restricted expression. *PADI1* is expressed in epidermis and uterus, *PADI3* in epidermis and hair follicles, *PADI4* in immune cells, brain, uterus and bone marrow, *PADI6* in ovary, egg cells, embryo and testicles [34]. *PADI2* is more widely expressed, including in the brain, uterus, spleen, breast, pancreas, skin, and skeletal muscles [3,6,34]. Their differential tissue expression patterns already suggest their unique and overlapping substrates in the genome. Human *PADIs* are homologous both within and between species (44−58% identity between human *PADI*s) [32]. Despite the high homology among family members, the five *PADI* enzymes have definite features, which facilitates the regulation of several distinct cellular processes. This feature may explain why altered levels of citrullination are associated with different pathological conditions, including multiple cancers [4,15,35,36,37,38,39,40,41]. Remarkably, *PADI2* and *PADI4* had distinct as well as overlapping substrates, for example, PADI4 directly citrullinates nuclear factor κB (NF-κB) p65 to promote its nuclear localization and transcriptional activity of key immune genes including IL-1β and tumor necrosis factor α (TNFα) to propagates inflammation in rheumatoid arthritis [42], whereas, mice with TNFα-induced inflammatory arthritis showed that *PADI2* contributes to TNFα-induced citrullination, but not required for NETosis. However, the authors had not investigated the PADI2 substrates in this mice model [43]. These studies highlight the relevance of citrullination to propagate inflammatory microenvironment, most likely also involved in cancer cells. Importantly, an elevated citrullination level associated with various cancers and neurodegeneration had been linked to increased release of exosomes and microvesicles (EMVs), which are lipid bilayer-enclosed structures released from cells and participate in cell-to-cell communication [44]. The citrullination connection with EMV release could be coupled with the increased calcium influx. Thus, it is imperative to examine the mechanisms by which citrullination modulates the EMV release during cancer progression. One cellular conditions linked to *PADI* family members is in response to hypoxia, characterized by insufficient blood supply. In malignant glioma cells hypoxia-induced increase of *PAD1, 2, 3*, and *4* mRNA levels depends on hypoxia-inducible factor-1α (HIF1α) [45]. Similarly, fibroblast-like synoviocyte hypoxia in humans predominantly induces the expression of PADIs through HIF1α regulation [46]. One unresolved question is how the hypoxic conditions in growing tumors influence *PADI* activity and citrullination.

## 3. Impact of *PADI* Family on Histone Citrullination

*PADIs* catalyzed histone citrullination also controls transcription regulation and chromatin organization [47,48,49,50,51,52,53,54,55,56]. Remarkably, citrullinated histones found to account for approximately 10% of all histone molecules, emphasizing the essential role of this posttranslational modification in many physiological processes in the nucleus [48]. For example, during preimplantation and early-stage embryo development histone citrullination can lead to chromatin changes [49]. Specifically, *PADI1* citrullinates both the histone H3 at 2, 8 and 17 positions (H3Cit2, 8, 17) and histone H4 at arginine 3 (H4Cit3) during 2- and 4-cell stage embryo development, which facilitates early embryo genome transactivation [50]. However, the potential implications of *PADI1*-directed histone citrullination had not been investigated yet. On the other hand, *PADI2* can specificity citrullinates histone H3 arginine 26 citrullination (H3Cit26) that leads to chromatin decondensation and transcriptional activation in human breast cancer cells [51,52]. *PADI2* can also catalyze H3Cit2, 8, 17 in mammary epithelial cells to regulate the expression of lactation related genes during diestrus [54]. PADI4 can generate H3Cit2, 8, 17 with higher efficiency than *PADI2* and also generate H4Cit3 and H3Cit2, 8, 17 to control chromatin organization and gene expression during key cellular processes [47,54]. Notably, PADI4-mediated citrullination of linker histone H1 regulates stem cell pluripotency during early embryogenesis [55,56].

Citrullination of histone H3 can promote oncogenesis by repressing the expression of tumor suppressor miRNAs, which ultimately increases oncogene mRNAs in somatolactotrope cells [57] (see Section 5). Histone citrullination modifies gene regulation by modulating chromatin condensation at several levels. For example, PADI4-mediated citrullination of histone H3 at arginine 8 (H3Cit8) leads to the eviction of heterochromatin protein 1 α (HP1α) from the chromatin at regulatory regions of human endogenous retroviruses (HERVs) and cytokines, which generates a chromatin accessible state and promotes their gene transcription [58]. These findings provided the first evidence that HP1α and citrullination are regulators of immune genes and HERVs, and that reactivation of cytokine genes and HERVs in multiple sclerosis patients arises from perturbed chromatin-mediated repression. Notably, citrullination of histone H1 promote its dissociation from DNA to create an open chromatin state that is essential for stem cell pluripotency during early embryogenesis [55,56]. Moreover, heterochromatin protein 1γ (HP1γ) citrullination within the chromodomain decreases its interaction with chromatin, which is necessary to facilitate the opening of chromatin in pluripotent stem cells [59]. However, the levels of histone H1 and HP1γ citrullination accompanied by compacted chromatin state during differentiation highlight the functional fine-tuning of citrullination to modulate the chromatin organization and gene expression.

*PADI4*-mediated histones citrullination also plays a critical role in response to DNA damage. Citrullinated histones interact with p53 through distinct domains and also associate with the p21 promoter to regulate gene expression of several p53 target genes [60]. Elevated levels of histone citrullination in hematopoietic cancer cell lines can induce apoptosis by increasing the expression of p53 and p21 [61]. Expression of the p53 target gene *OKL38* gene that encodes an oxidative stress response protein, is induced by DNA damage and regulated by H3Cit2, 8, 17 [62]. The tumor suppressor protein, inhibitor of growth 4 (ING4), undergoes citrullination and regulates gene expression. citrullination of ING4 at the nuclear localization sequence prevents p53 binding to ING4, represses p53 acetylation, and consequently inhibits downstream p21 expression [63]. Further, citrullination of histone chaperone protein, nucleophosmin can occur in p53 dependent manner, which promotes its translocation from the nucleoli to the nucleoplasm and controls the p53-driven growth [64]. DNA damage via the p53 pathway also induces H4Cit3 and of Lamin C around fragmented nuclei in apoptotic cells. Since H4Cit3 negatively correlates with p53 protein expression and with tumor size in non-small cell lung cancer tissues, it is proposed to play a crucial role in carcinogenesis [65]. Taken together, these findings suggest citrullination in the p53 signaling pathway and DNA damage contribute to altered gene regulation in cancer.

## 4. PADI2 Shows Specific Substrate Preferences

*PADI2* is the most widely expressed member of the *PADI* family. Compiling the data for all *PADI* gene family members from RNA-sequencing experiments across 106 human tissues (Gene expression Atlas, https://www.ebi.ac.uk/gxa/genes/ensg00000117115, Figure 2B), including NIH Roadmap Epigenomics Mapping Consortium, Encyclopedia of DNA Elements (ENCODE), Genotype-Tissue Expression (GTEx) Project, and Illumina body map project [66,67,68,69,70], shows that *PADI2* is highly and widely expressed. Phylogenetic analysis of the *PADI* family showed that *PADI2* is the most conserved family member and likely the founder member of the family. Interestingly, *PADI1* and *PADI3* enzymes, as well as PADI4 and PADI6 enzymes are more closely related to each other than to *PADI2* [8].

One unique feature of *PADI2* is that arginine catalysis relies on a substrate-assisted mechanism, so sequential ordered calcium binding is required for proper positioning of a catalytic cysteine-645 residue in relation to the substrate pocket [32,71,72]. This mechanism confers selectivity for certain substrates leading to specific citrullination of H3Cit26 [50,51,52]. Interestingly, H3Cit26 preferentially associates with SWI/SNF-related matrix-associated actin-dependent regulator of chromatin subfamily A containing DEAD/H box 1 (SMARCAD1), which regulates naive pluripotency by suppressing heterochromatin formation [73]. This study also supports our previous finding that histone citrullination weakens the binding of H3K9me3 (lysine 9 trimethylation of histone H3) to the heterochromatin proteins, which consequently leads to gene activation of key transcription units [58,73]. Combined, this evidence supports the notion that *PADI2* exhibits a unique ability to citrullinates specific proteins that regulate gene expression. Moreover, *PADI2* specifically citrullinates glial fibrillary acidic protein (GFAP), an astrocyte-specific marker protein found in Alzheimer’s disease (AD) patients [74]. This work demonstrates the clinical relevance of one specific *PADI2* substrate in AD patients. However, the mechanistic insights of this GFAP citrullination remain unaddressed.

Specifically, *PADI2* is the most prevalent expressed *PADI* member and is preferentially expressed in oligodendrocytes (OLs) and other glial cells [75] that link *PADI2* specific actions to myelination. The most prominent functions of OLs is forming the myelin sheath, which is essential for proper neuronal communication and central nervous system function [76]. Notably, *PADI2* levels increased during OLs differentiation, and PADI2 acts as a chromatin modifier to promotes oligodendrocyte precursor cell differentiation [77]. Therefore, *PADI2* contributes to efficient myelination and motor and cognitive functions.

*PADI2* is highly expressed in peripheral blood mononuclear cells of healthy individuals and was also found to affect the differentiation of T helper (Th) cells, which play a critical role in immune defense and are associated with autoimmune and allergic diseases [78]. Further, *PADI2* specifically citrullinates arginine330 of GATA3 and arginine4 of RORtγ, key transcription factors required for Th cell functions. These are *PADI2* specific substrates that are not citrullinated by *PADI4*, which further highlights the unique features of *PADI2*. Interestingly, GATA3 citrullination weakens its DNA binding, whereas RORtγ citrullination strengthens its DNA binding ability, so PADI2-directed citrullination attenuates the Th2 cells differentiation and potentiates Th17 cells differentiation [9]. Taken together, these results highlight the functional implication of PADI2 in immune regulation and Th cell driven diseases.

## 5. The Role of *PADI2* in Cancer

Citrullination is also associated with multiple cancers (see [41] for a review on citrullination in cancer and [79] for a review on citrullination in gastrointestinal cancers). Given the fact that *PADI2* and 4 are the most expressed in humans, therefore most of the studies had focused on understanding their functions and other members of the *PADI* family have not been studied in the context of their implications in the specific type of cancer progression. More detailed function of *PADI4* in cancer had been described previously [41]. Among *PADI* family members, *PADI2* is highly expressed in primary breast tumors and luminal breast cancer cell lines [80,81]. *PADI2* regulates transcription in human mammary epithelial cells [82,83], promotes epithelial-to-mesenchymal transition in mammary tumor cells [84], and facilitates ductal invasion in primary mouse mammary organoids [84]. Meta-analysis from large cohorts of breast cancer patients and related healthy individuals showed that among *PADI* gene family members, only *PADI2* is overexpressed in breast cancer and other cancers, as its overexpression correlates with poor prognosis [85]. All these observations clearly highlight the functional association of *PADI2* in breast cancer pathogenesis.

In addition to its role in inflammation and in breast cancers, *PADI2* is also implicated in other cancer types, including skin cancer. *PADI2* overexpressing transgenic mice spontaneously develop neoplastic skin lesions, of which a subset can progress to invasive squamous cell carcinoma (SCC). These lesions display an increased level of invasiveness and epithelial-to-mesenchymal transition, along with a high level of inflammation [86,87]. These studies suggest that *PADI2* facilitates skin tumor progression by enhancing inflammation within the tumor microenvironment. In this context, further mechanistic and clinical studies will reveal how using *PADI2* based screening assays will improve cancer treatment strategies.

In castration-resistance prostate cancer (CRPC) patients, *PADI2* is overexpressed and modulates androgen receptor (AR) signaling despite androgen depletion therapy (88). *PADI2* gene expression is required for cell-cycle progression of prostate cancer cells and for proliferation of CRPC prostate cancer cells both in vitro or in vivo. *PADI2*-directed citrullination stabilizes the AR and mediates its nuclear translocation, which reduces AR degradation and facilitates its binding to target genes [88]. Taken together, *PADI2* promotes prostate cancer progression. Further, *PADI2* overexpression can occur in multiple cancers, including breast ductal carcinoma, ovarian serous papillary adenocarcinoma, cervical squamous cell carcinoma, liver hepatocellular carcinoma, lung cancer, and also papillary thyroid carcinoma [89]. Finally, *PADI2* may mediate an essential function in gastric cancers by promoting cell proliferation and invasion [79,88].

In contrast to these cancers, *PADI2* levels can be downregulated in colorectal cancer (CRC) patients. This was first time reported by Cantariño et al. [90] as they found low levels of PADI2 expression in 98 CRC patients compared to adjacent mucosa from 50 healthy controls. Importantly, this downregulated of *PADI2* expression in CRC patients correlated with poor prognosis [90]. Likewise, *PADI2* is required to inhibit cellular proliferation of colon cancer cells by arresting the G1 phase of the cell cycle [91]. This result likely arose through *PADI2*-mediated citrullination which plays a critical role in Wnt (wingless)/ β-catenin signaling pathway in CRC patients [92]. In fact, the anti-parasitic nitazoxanide used to treat CRC patients, efficiently inhibits the Wnt signaling pathway by increasing the stability of the PADI2 protein and enhancing the citrullination of β-catenin, which inhibits the Wnt pathway and thereby decrease the cellular proliferation of colon cancer cells.

Notably, *PADI2* is also found to be highly expressed in bone marrow mesenchymal stem cells (BMMSCs) [93]. These BMMSCs are key components for both the normal and transformed bone marrow niche. The stable alteration in the bone marrow niche leads to multiple myeloma, an incurable malignancy of clonal plasma cells, which accounts for approximately 10% of hematological malignancies [94]. PADI2 can increase interleukin-6 (IL-6) levels by modulating occupancy of the histone H3Cit26 mark at the regulatory region of IL-6 by malignant plasma cells in response to the chemotherapeutic agent, bortezomib which suggests *PADI2* is involved in the acquisition of drug resistance [93,95]. Importantly, monitoring of *PADI2* enzymatic activity is proposed in the early stage of multiple myeloma patients to improve the efficacy of their therapeutic strategies [95].

## 6. PADI2 Is a Novel Regulator of Transcription

The regulated transcription of genes determines cell identity and malignant transformation and citrullination influences gene expression [30,70]. However, how *PADI2* influences gene expression in cancer cells is not fully understood. Recently, we have identified a novel *PADI2* specific substrate central to the basal transcriptional machinery. We found that *PADI2*, but not the other *PADI* family members, citrullinates arginine1810 (Cit1810) in repeat 31 of the carboxyl-terminal domain (CTD) of the largest subunit of RNA polymerase II (RNAP2) [85]. Strikingly, cit1810 is crucial for RNAP2 to overcome the pausing barrier close to the transcription start site, which enable the efficient transcription of highly expressed genes needed for cell cycle progression, metabolism, and cell proliferation [85]. So, we developed a specific antibody that recognizes Cit1810 and discovered this modification occurs in active RNAP2 phosphorylated at serine residues of the conserved CTD repeats [85].

However, in vitro citrullination of R1810 by PADI2 is blocked by phosphorylation of the adjacent residues, suggesting that repeat 31 is not phosphorylated in vivo [85]. In fact, cit1810 enhances the association of active RNAP2 with the positive transcription elongation factor b (P-TEFb) kinase complex, comprises of cyclin dependent kinase 9 (CDK9) and Cyclin T1 (CCNT1). Thus, cit1810 enables paused RNAP2 to overcome pausing and facilitates gene transcription needed for cellular growth [96]. Taken together, we proposed that PADI2-mediated cit1810 of RNP2 enhances the interaction of the CTD with P-TEFb complex and its associated factors including SEC (super elongating complex), which facilitates the productive elongation (Figure 3). The citrullination of R1810 is required for RNAP2 pause release or for high turnover at promoters of highly expressed genes that maintain cellular proliferation [97,98,99], which may help drive tumor progression and metastasis. This observation is consistent with the recent findings that inhibiting of PADI2 activity suppresses the mammary gland tumor invasion in mice [83] and reduces the mammary cancer progression in dogs and cats [100]. Taken together, this work highlights a mechanism by which PADI2 overexpression may contribute to tumorigenesis.

Since no enzymes that erase citrullination have been described, alternative mechanisms may exit to replace the citrullinated RNAP2 before termination of the transcription cycle. Importantly, as observed for other dynamically regulated arginine residues [7,101], R1810 at RNAP2-CTD can also undergo asymmetrical and symmetrical dimethylation by coactivator associated arginine methyltransferase 1 (CARM1) and protein arginine methyltransferase 5 (PRMT5), respectively [102,103]. Notably, the asymmetrical dimethylation (me2a) of R1810 occurs mainly in hypo-phosphorylated RNAP2 or prior to the initiation of transcription. R1810 is involved in regulating the expression of the small nuclear RNAs (snRNAs) and the nucleolar RNA (snoRNA) in human cells [102]. However, PRMT5 catalyzed symmetrical dimethylation (me2s) of R1810 promotes the recruiting of the survival of motor neuron protein and its interaction with senataxin, a helicase that enhances transcriptional termination [103]. We found that PADI2 cannot use me2a R1810 as a substrate and that citrullination precludes PRMT5 from catalyzing me2s R1810. Moreover, depletion of PRMT5 and/or CARM1 does not influence PADI2 mediated citrullination of R1810. Such depletion also does not interfere with its transcriptional effect nor with the interaction of citrullinated RNAP2 and the P-TEFb complex. These results indicate that R1810 methylation and citrullination are independent regulatory pathways [85].

Future studies should elucidate how a relative minor change of a single arginine to citrulline in the middle of CTD alters cell fate. The loss of the positive charge of the arginine residue could affect the structure by directly changing electrostatic interactions with relevant proteins like P-TEFb. However, arginine-mediated interactions of the intriniscally disordered protein domain of RNAP2-CTD with RNAs and its associated proteins may participate in conjunction with serine residues phosphorylation within consensus CTD repeats to induce changes in macromolecular condesates within the cell nucleus [104,105,106]. In that respect, two independent macromolecular condensates have been postulated, one related to transcription initiation or mediator complex condensate and another with transcription elongation and splicing factors [107]. Recently elongating form of RNAP2 was found to be preferentially incoporated into condesates that are formed by the splicing factors [107,108]. In fact, inhibition of PADI2 activity or mutation of R1810 produces large changes in RNA splicing (unpublished observation). Based on these results, we posit that cit1810 at RNAP2 could participate in modulating these key interactions, which can consequently influence transcriptional output and RNA processing through liquid-liquid phase separation (LLPS, Figure 3). Further work will resolve these possibilities.

PADI2 plays also an essential role in mediating the activation of estrogen receptor (ER) target genes by citrullination of histone H3 arginine 26 (H3Cit26) on chromatin. In this case, PADI2 facilitates transcriptional activation by modulating the chromatin architecture around the estrogen receptor elements of estradiol-induced genes [51]. Further, two minutes of estradiol exposure increased H3Cit26 at the ERα binding sites in chromatin, suggesting that PADI2 mediated H3Cit26 facilitates ERα access to DNA [52]. Given that progestin activated progesterone receptor modulates gene expression and genome organization of breast cancer cells by targeting kinases and chromatin remodelers to chromatin [109,110,111], we are investigating whether PADI2 directed citrullination can modulate progestin gene regulation and genome organization in breast cancer cells.

Many elongation factors and kinases participate in controlling the RNAP2 transcription pause release, a mechanism that regulates the expression of many genes involved in cancer progression and metastasis, like cyclin dependent kinase 9 (CDK9), v-myc myelocytomatosis viral oncogene homolog (MYC), Jumonji domain containing 6 (JMJD6), arginine demethylase and lysine hydroxylase and BRD4 [112,113,114,115,116]. We found that the absence of PADI2-mediated cit1810 of RNAP2 reduced cell proliferation of breast cancer cells, by modulating cell cycle progression. PADI2 modifies the transcriptional machinery and promotes the oncogenic gene regulation. PADI2 depletion drastically affects the transcription elongation of key genes, including *MYC* gene, which eventually inhibits cell proliferation [85]. Given the tight link between transcription elongation and RNA splicing [117], it will intriguing to explore how citrullination regulates RNA processing during tumor progression.

## 7. Conclusions and Perspectives

Protein arginine citrullination affects multiple cellular processes by adding a new functional group that modulates key interactions with other proteins or with RNA. Despite the relevance of citrullination in multiple pathophysiological conditions, our knowledge of the underlying molecular mechanisms remains quite limited, so we must overcome this barrier to develop novel therapeutic avenues. PADI2 is the most widely expressed isoform of the PADI family and is highly prevalent across multiple tumor tissues, indicating its potential functional relevance in tumorigenesis. In this review, we have summarized PADI2 specific functions on transcription regulation not only by citrullination of arginine 26 on histone H3 but also by targeting the master regulator of transcription, namely RNAP2. Noteworthy, cit1810 of RNAP2 controls global transcription by modulating the recruitment of P-TEFb and associated protein complexes to the transcription machinery. We highlight the utility to elucidate how PADI2-mediated citrullination contributes to cancer metastasis within the specific tumor microenvironment. We propose PADI2 is a novel therapeutic target, not only for autoimmune and inflammatory diseases, but also for several cancer types.

## Figures and Tables

**Figure 1 ijms-21-01351-f001:**
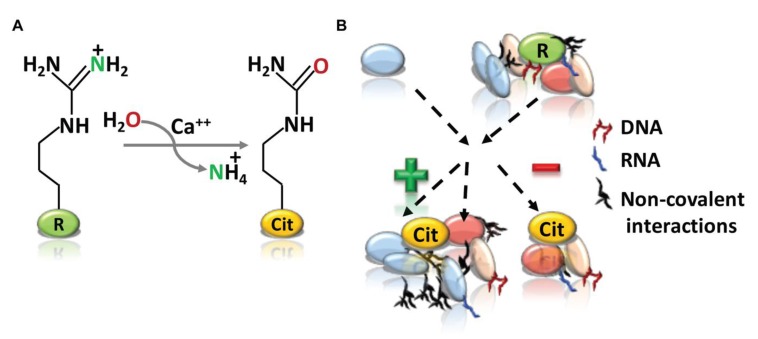
Arginine citrullination and its cellular functions. (**A**) The peptidyl arginine deiminases (PADI) enzyme catalyze peptidyl-arginine (positively charged) to peptidyl-citrulline (neutral in charge) and release of ammonia. (**B**) Arginine (green) to citrulline (orange) conversion could consequently affects the functional protein-protein and protein-nucleic acid interactions by affecting the protein folding, as well as intermolecular and intramolecular interactions. “+” and “−” signs representing the positive and negative interactions respectively.

**Figure 2 ijms-21-01351-f002:**
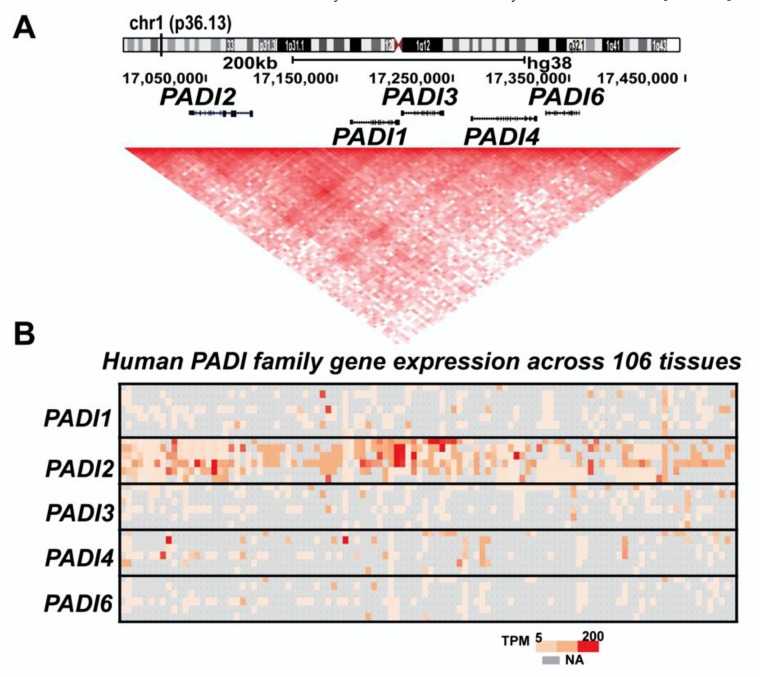
Human *PADI* gene family. (**A**). Five human *PADIs* genes (*PADI1*, *PADI2*, *PADI3*, *PADI4* and *PADI6*) are present in the p36.13 of chromosome 1. Among the family members, *PADI2* gene is the only one in reverse direction. *PADI* loci encompasses the same topological associated domain (TAD) in T47D breast cancer cells. (**B**). *PADI2* is widely expressed among family members. Heatmap representing the basal expression level of all five human *PADI* members across 106 human tissues compiled from the gene expression atlas (https://www.ebi.ac.uk/gxa/home). Basal gene expression represented as TPM, transcripts per million from RNA-sequencing.

**Figure 3 ijms-21-01351-f003:**
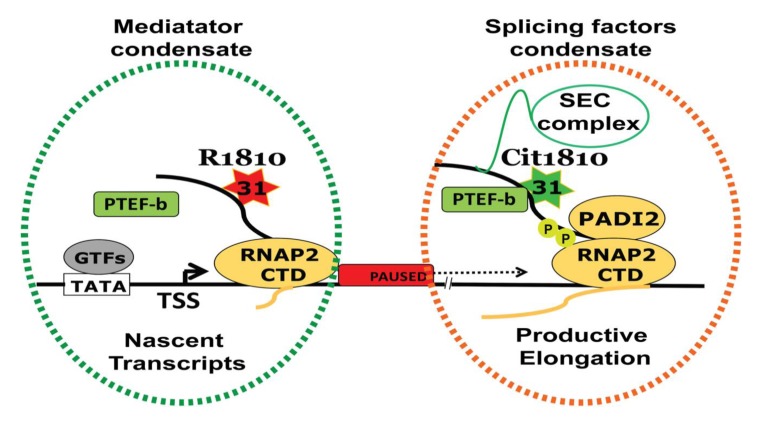
PADI2-mediated cit1810 RNAP2 is a new player in transcription regulation. PADI2 catalyzed R1810 to Cit1810 at RNAP2-CTD facilitates the association with the P-TEFb complex (comprises CDK9-CCNT1). This interaction overcomes RNAP2 pausing and increases in transcription and cell proliferation. An increased association between the P-TEFb complex and RNAP2-CTD most likely attracts associated complexes including Super Elongating Complex (SEC) to facilitate active RNAP2 encompassing splicing factor condensate (orange dotted) derive transcription regulation. However, R1810 along with nascent transcripts forming mediator complexes condensate (green dotted). TSS = transcription start site; GTFs = general transcription factors.

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
