# Peer review of "Peptidyl Arginine Deiminase 2 (PADI2)-Mediated Arginine Citrullination Modulates Transcription in Cancer"

_ijms, 2020, doi:10.3390/ijms21041351_

Round 1
Reviewer 1 Report
In this manuscript, M Beato and P Sharma report about recent progress, including the author's own works, in the role of peptidylarginine deiminase 2 (PADI2) in modulation of transcription and possible involvement in cancer. The manuscript is well constructed, informative, and understandable for readers even with little background. But I have some enquiries for, I think, improvement of the manuscript, as follows.
Please add a paragraph discussing more precisely the similitudes/differences between PADI2 and PADI4 implication in cancer and gene expression control. Especially in the Introduction, “Histone citrullination …” paragraphs, what is the respective role of each PADI? It would be nice to discuss how PADI2 is translocated to the nucleus since it does not display a nuclear localization signal. There are too many abbreviations, this makes difficult the reading. Please use an abbreviation only when a word is used more than three times all along the manuscript. Please, in the abbreviation table, please range the abbreviation in alphabetical order. There are many spelling errors, a careful reading of the manuscript is necessary. See the enclosed file for some examples. Line 42; filaggrin is not degraded by the proteasome as stated but by several proteases including caspase 14 and others. See the recent review in Int. J. Mol. Sci. by Méchin et al 2020. This review could be cited in addition to the old paper by Senshu et al [10]. Please clarify the following sentences: lines 81-83 (“These findings provided the first evidence that HP1α and citrullination are regulators of immune genes and HERVs, and that reactivation of cytokine genes and HERVs in multiple sclerosis patients arises from a deficient chromatin-mediated repression” is this the correct meaning of your sentence?). Line 239, “Importantly, PADI2 based screening is proposed in early stage of….”. The two last sentences of Figure 3 legend. Line 302, a word is apparently missing in the following sentence “and those influencing transcritional output and RNA processing through liquid-302 liquid phase separation (LLPS)”. On figure 1, what is the significance of the black lines similar to the red ones / DNA and blue ones / RNA? Reference 77 is now published in Mol Cell 2019, 73, 84-96. Line 95, please specify which arginine is citrullinated if known. By the way, please use only one way to note the citrullinated arginine, either H3cit8 or cit-H4R3 or cit26H3. Line 120 and figure 2 legend, please change to “topologically associating domain” and add a reference.
Author Response
We thank all the reviewers for their useful suggestions. In order to answer the specific points raised, we will use the following code in the point-by-point answers below:
Original comments from the reviewers are in black.
Our answers to the comments are depicted in blue.
Changes in the manuscript as compared to the initial version are shown in red.
Reviewer 1
In this manuscript, M Beato and P Sharma report about recent progress, including the author's own works, in the role of peptidylarginine deiminase 2 (PADI2) in modulation of transcription and possible involvement in cancer. The manuscript is well constructed, informative, and understandable for readers even with little background. But I have some enquiries for, I think, improvement of the manuscript, as follows.
Point 1: Please add a paragraph discussing more precisely the similitudes/differences between PADI2 and PADI4 implication in cancer and gene expression control. Especially in the Introduction, “Histone citrullination …” paragraphs, what is the respective role of each PADI? It would be nice to discuss how PADI2 is translocated to the nucleus since it does not display a nuclear localization signal.
Response 1: We acknowledge the reviewer point.
As suggested by reviewer 3, we introduced a new section entitled ¨ Impact of PADI family on Histone citrullination¨ (section 3) after the brief description of PADI family members (section 2). In this section, we discussed the respective role of PADI family members on histone citrullination.
In section 2, we explained how PADI2 is translocated to the nucleus and hence cited the recent publication from Thompson´s group.
Similarities/differences between PADI2 and PADI4 and their potential implications in cancer had been previously discussed in the review by (Yuzhalin, A.E. Citrullination in Cancer. Cancer Research 2019,79,1274-1284). However, we agreed with a reviewer to highlight the significance of both these enzymes here and therefore in both sections 2 and 3, we discuss overlapping and specific targets of these enzymes and their potential implication in gene expression and cancer.
Point 2: There are too many abbreviations, this makes difficult the reading. Please use an abbreviation only when a word is used more than three times all along the manuscript. Please, in the abbreviation table, please range the abbreviation in alphabetical order. There are many spelling errors, a careful reading of the manuscript is necessary. See the enclosed file for some examples. Line 42; filaggrin is not degraded by the proteasome as stated but by several proteases including caspase 14 and others. See the recent review in Int. J. Mol. Sci. by Méchin et al 2020. This review could be cited in addition to the old paper by Senshu et al [10].
Response 2:
The abbreviation table had been organized in alphabetical order. Following the reviewer´s suggestions, we only included abbreviations in the table if a word is used more than three times all along with the manuscript. All the spelling errors had been corrected in the attached file. Regarding line 42-citrullination of filaggrin (Filament Aggregating protein that binds to keratin fibers in epithelial cells) leads to the partial unfolding of this protein, decreases its200209 affinity for keratins, and, thereby, makes it more prone to degradation by several proteases including caspase 14 [10;11] and hence Méchin et al 2020 had been cited here.
Point 3: Please clarify the following sentences: lines 81-83 (“These findings provided the first evidence that HP1α and citrullination are regulators of immune genes and HERVs, and that reactivation of cytokine genes and HERVs in multiple sclerosis patients arises from a deficient chromatin-mediated repression” is this the correct meaning of your sentence?).
Response 3:
This is the correct meaning and now sentence had been modified to ¨These findings provided the first evidence that HP1α and citrullination are regulators of immune genes and HERVs, and that reactivation of cytokine genes and HERVs in multiple sclerosis patients arises from perturbed chromatin-mediated repression¨.
Point 4: Line 239, “Importantly, PADI2 based screening is proposed in early stage of….”.
Response 4: This sentence had been corrected to ¨ Importantly, monitoring of PADI2 enzymatic activity is proposed in the early stage of multiple myeloma patients to improve the efficacy of their therapeutic strategies¨.
Point 5: The two last sentences of Figure 3 legend. Line 302, a word is apparently missing in the following sentence “and those influencing transcriptional output and RNA processing through liquid-302 liquid phase separation (LLPS)”.
Response 5: These sentences present in the manuscript text and now corrected to ¨ Based on these results, we posit that cit1810 at RNAP2 could participate in modulating these key interactions, which can consequently influence transcriptional output and RNA processing through liquid-liquid phase separation (LLPS) (Figure 3) ¨.
Point 6: On figure 1, what is the significance of the black lines similar to the red ones / DNA and blue ones / RNA?
Response 6: Here, a black line representing the non-covalent interactions to maintain the protein-protein interactions and this had been corrected in figure 1.
Point 7: Reference 77 is now published in Mol Cell 2019, 73, 84-96. Line 95, please specify which arginine is citrullinated if known. By the way, please use only one way to note the citrullinated arginine, either H3cit8 or cit-H4R3 or cit26H3. Line 120 and figure 2 legend, please change to “topologically associating domain” and add a reference.
Response 7: All the above-mentioned corrections had been considered.
We only cited Molecular Cell paper.
Line 95- this was H3Cit2,8,17, One way had been followed to represent citrullinates arginine residues.
In figure 2 legend-correction had been included and our reference had been included (Le Dily et al. 2014).
Reviewer 2 Report
This is a nicely written review of PAD2. The main focus of this review is to discuss the role of PAD2 in transcriptional regulation in the context of cancer. The authors have nicely summarized most of the PAD2 related literature available to date. Minor comments: - A typo in the main title, 'deiminases' is actually 'deiminase'. - To get a complete picture of PAD2 and its role, authors should also comment on its significance in Rheumatoid Arthritis, similarities or differences to PAD4, role in NF-kB citrullination. You may cite and discuss the following papers in relevant sections in this review, PMID: 28783661 (citrullination of NF-kB), 28188029 (PAD2 vs PAD4 citrullination in mouse models), 30161253 (relative efficiencies of PAD2 vs PAD4)
Author Response
We thank all the reviewers for their useful suggestions. In order to answer the specific points raised, we will use the following code in the point-by-point answers below:
Original comments from the reviewers are in black.
Our answers to the comments are depicted in blue.
Changes in the manuscript as compared to the initial version are shown in red.
Reviewer 2
Comments and Suggestions for Authors
This is a nicely written review of PAD2. The main focus of this review is to discuss the role of PAD2 in transcriptional regulation in the context of cancer. The authors have nicely summarized most of the PAD2 related literature available to date.
Minor comments:
Point 1: A typo in the main title, 'deiminases' is actually 'deiminase'.
Response 1: We thank the reviewer for his/her constructive suggestions. The typo mistake in the main title has been corrected.
Point 2: To get a complete picture of PAD2 and its role, authors should also comment on its significance in Rheumatoid Arthritis, similarities or differences to PAD4, role in NF-kB citrullination. You may cite and discuss the following papers in relevant sections in this review, PMID: 28783661 (citrullination of NF-kB), 28188029 (PAD2 vs PAD4 citrullination in mouse models), 30161253 (relative efficiencies of PAD2 vs PAD4).
Response 2: As pointed by a reviewer the main focus of this review is to discuss the role of PADI2 in transcriptional regulation in the context of cancer. Here, the significance of PADI2 in Rheumatoid Arthritis or other autoimmune diseases will be out of scope.
However, we thank a reviewer for positive suggestion and hence we discussed NF-kB citrullination (PMID: 28783661), PADI2 vs PADI4 in the mouse model (PMID: 28188029), as well as relative efficiencies of PADI2 vs PADI4 (PMID: 30161253) in section 2 encompassing ¨Functional role of PADI family¨.
Reviewer 3 Report
The thesis received for review concerns important issues both from the cognitive and application point of view. In my opinion, its subject matter is in line with the profile of the magazine Int J Mol Sci.
In my opinion, minor shortcomings concern the structure of the manuscript. The third
paragraph of the subsection "Introduction" is detailed and should be
distinguished in a separate section (eg. under the title "Role of PADI
family in control of transcription regulation via histone citrullination").
A good idea is to separate the section titled "Abbreviations". However, not all explanations used were explained, eg. "HEK" (line 32); RGG / RG (line 34); IL-1 (line 67 - only IL-6 has been explained later); H3K9me3 (line 168); RNAP2 (line 247); CARM1 and PRMT5 (line 277). In addition, the PADI script (line 110) was explained earlier (line 30).
In addition, the following errors and weaknesses should be corrected before submitting the paper for printing:
The citation [37] should be removed from line 109 and inserted after the authors' names (line 108). Called enzymes from line 110 should start with a lowercase letter, and the spelling of the word "DeIminases" should be corrected to "deiminases". The word "metal" (line 114) should be removed. The name cysteine (line 119) should start with a lowercase letter. The script "PAD2" (line 180) should be corrected to "PADI2". After the authors' names (line 222) a citation number should be given [82]. The spelling of the script "p-TNFb" should be the same in lines 291 and 353.
Author Response
We thank all the reviewers for their useful suggestions. In order to answer the specific points raised, we will use the following code in the point-by-point answers below:
Original comments from the reviewers are in black.
Our answers to the comments are depicted in blue.
Changes in the manuscript as compared to the initial version are shown in red.
Reviewer 3
The thesis received for review concerns important issues both from the cognitive and application point of view. In my opinion, its subject matter is in line with the profile of the magazine Int J Mol Sci.
Point 1: In my opinion, minor shortcomings concern the structure of the manuscript. The third paragraph of the subsection "Introduction" is detailed and should be distinguished in a separate section (eg. under the title "Role of PADI family in control of transcription regulation via histone citrullination").
Response 1: We agreed with a reviewer´s suggestion. Therefore, we introduced a new section entitled ¨ Impact of PADI family on Histone citrullination¨ (section 3) after the brief description of PADI family members (section 2). Additionally, in this section, we discussed the respective role of PADI family members in histone citrullination.
Point 2: A good idea is to separate the section titled "Abbreviations". However, not all explanations used were explained, eg. "HEK" (line 32); RGG / RG (line 34); IL-1 (line 67 - only IL-6 has been explained later); H3K9me3 (line 168); RNAP2 (line 247); CARM1 and PRMT5 (line 277). In addition, the PADI script (line 110) was explained earlier (line 30).
Response 2: All the suggested explanations had been added to the manuscript.
Point 3: In addition, the following errors and weaknesses should be corrected before submitting the paper for printing: The citation [37] should be removed from line 109 and inserted after the authors' names (line 108). Called enzymes from line 110 should start with a lowercase letter, and the spelling of the word "DeIminases" should be corrected to "deiminases". The word "metal" (line 114) should be removed. The name cysteine (line 119) should start with a lowercase letter. The script "PAD2" (line 180) should be corrected to "PADI2". After the authors' names (line 222) a citation number should be given [82]. The spelling of the script "p-TNFb" should be the same in lines 291 and 353.
Response 3: We thank a reviewer for all the positive suggestions here. All the mentioned corrections had been considered here.
-We inserted respective citations after the author´s name.
- Spelling mistakes had been corrected in the previous line 110.
- The word ¨Metal¨ had been removed.
- The word cysteine, PADI2, and P-TEFb had been corrected.